# Applying Ant Colony Optimization to Reduce Tram Journey Times

**DOI:** 10.3390/s24196226

**Published:** 2024-09-26

**Authors:** Mariusz Korzeń, Igor Gisterek

**Affiliations:** Department of Civil Engineering, Wrocław University of Science and Technology (Politechnika Wrocławska), Wybrzeże Wyspiańskiego 27, 50-370 Wrocław, Poland; igor.gisterek@pwr.edu.pl

**Keywords:** nature-inspired algorithm, ant colony optimization, public transport, drive time, route optimization, trams

## Abstract

Nature-inspired algorithms allow us to solve many problems related to the search for optimal solutions. One such issue is the problem of searching for optimal routes. In this paper, ant colony optimization is used to search for optimal tram routes. Ant colony optimization is a method inspired by the behavior of ants in nature, which as a group are able to successfully find optimal routes from the nest to food. The aim of this paper is to present a practical application of the algorithm as a tool for public transport network planning. In urban public transport, travel time is crucial. It is a major factor in passengers’ choice of transport mode. Therefore, in this paper, the objective function determining the operation of the algorithm is driving time. Scheduled time, real time and theoretical time are analyzed and compared. The routes are then compared with each other in order to select the optimal solution. A case study involving one of the largest tramway networks in Poland demonstrates the effectiveness of the nature-inspired algorithm. The obtained results allow route optimization by selecting the route with the shortest travel time. Thus, the development of the entire network is also possible. In addition, due to its versatility, the method can be applied to various modes of transport.

## 1. Introduction

Of the public transport modes used in cities, rail transport offers many advantages. But not every city has an extensive rail junction or subway network. As transport systems become increasingly complex, their infrastructure and vehicles put significant financial pressure on local authority budgets. As a result, not only do new systems need to be monitored for effectiveness [1], but the same refers to operating, renewing and expanding the existing ones. Being a reasonable compromise between capacity and speed offered by commuter railways and subways, and the lower purchase price of buses, various forms of light rail vehicles (LRVs) or trams are used in many cities. A significant advantage of tram transport is its high capacity with reasonable investment and operating costs, coupled with efficiency and reliability [2,3,4].

Even in fully separated systems, it is not easy to optimize the traffic. This is due to many factors which may oppose each other. As collected and summarized by Singh et al. [5], in-depth research has been conducted on the components of total travel time, including platform wait times, on-train times and train dwell times. Jafarian-Moghaddam, in his paper [6], cites more than 20 researchers who have conducted scientific work on this subject, dividing the desired features in train scheduling into the following objectives: energy consumption and travel time, as well as constraints and variables such as tractive force, resistance force, rail route conditions and economical speed. Some of the factors mentioned vary within given limits, similar to many systems across the world. For instance, the value of acceleration and deceleration of vehicles is limited by two decisive factors, namely friction between wheels and rails, and the ability of standing passengers to remain in vertical position. Other factors may vary significantly; a 4.5 km long tram route through center of Warsaw, Poland, between Zawisza Square and Washington Square, is almost perfectly straight, has no turnouts except for a balloon loop (Starynkiewicza) and is crossed by only two other tram lines, at Central Station and along Marszałkowska Street. The same distance across the center of Wrocław, Poland, measured between John Paul II Square and Szczytnicki Bridge, takes no less than seven turns, is crossed by six other tram routes and includes almost 20 turnouts. Both of these networks are shown in Figure 1.

This simple comparison shows that networks designed in similar period, according to the same regulations, in the same country and having a similar cross-center character may look completely different. The conditions mentioned above have a decisive character concerning total effectiveness and the capacity of the network. A vast number of tram tracks, car lanes, and pedestrian and bike routes crossing a mass transit corridor on the same level must be included in any optimization of the network or capacity analysis as they increase the variability of the estimated travel time between two adjacent stops. The case is further complicated because railway vehicles are also influenced by variable passenger exchange time. In some cases, one should expect extended times of boarding and alighting (morning and afternoon peak hour, bank holidays, mass events, etc.), while others are mostly accidental and hard to predict; for instance, when a passenger of reduced mobility (PRM) requires assistance with boarding using vehicle ramp.

Another key factor having influence on the system capacity and operation speed is its complexity, understood as a possibility to choose more than one route to connect two points. Under specific conditions this may also be defined as the redundancy of the network. In developed systems, the complexity of the connections may be further expanded by taking a route which requires an interchange. For practical and economic reasons (maintaining attractive headway), networks which operate on four or more termini rely on interchanges. A system heavily or even fully relying on interchanges is typical for subway systems and recently built light rail networks (‘one route, one line’ philosophy), as opposed to commuter railways, especially S-Bahn systems in Germany with their trunk lines (Stammstrecke) and heritage tram systems [7]. Networks of tramways in Strasbourg, following the mentioned ‘one route, one line’ growth pattern, is shown in Figure 2. Events like use of a ramp by a PRM, as mentioned in previous paragraph, often cause delays on other lines using the same corridor and influence perpendicular mass transit corridors if there is a level crossing. In conclusion, the more complex a network is, the harder it is to propose connections attractive for the passengers based only on non-scientific, traditional methods and precisely predict arrival times at given stations.

This leads towards the necessity to optimize the network. It is a multidisciplinary task which may be handled using various approaches. Xiang-Ming et al. [8], in their paper, used a multi-agent approach to simulate and evaluate an urban rail transit network. They concentrated on the spatial and temporal characteristics of passenger flow distribution. The route choice model used in the paper mentioned is based on Logit. The model was validated using real passenger flow data from Beijing subway and therefore limited for comparison, as this railway system is fully separated from other forms of traffic, as opposed to typical tramway systems. Landex and Nielsen [9] utilized a passenger regularity model and differentiated passenger delays and train delays based on the case of the Copenhagen metro. Again, the object of the research was a fully separated system. In a more recent paper, Hickish et al. [10] examine the applicability of a Bayesian algorithm to maximize passenger satisfaction and benchmark it against a genetic algorithm method. Xu et al. [11] used the same method to optimize passenger satisfaction via improved timetabling, similarly to Wei and Yuan [12]. However, the cases mentioned above see the only constraint as the overcrowding of vehicles during rush hours, since these papers describe fully separated subway systems. This causes only partial applicability of methods described for the research on a complex tram network.

Heuristic methods are used to solve many transport problems. In [13], the authors used a heuristic method to design a transit route network, taking into account a number of important parameters such as budget, service level standards and the attractiveness of transit routes. In [14], a genetic algorithm was used to investigate the basic features of the optimal bus route network design problem (BTRNDP) with variable travel demand. In [15], a genetic algorithm was also used, but to implement an adaptive network to optimize public transport routes by reducing the available street network. There are also other attempts to optimize entire transit networks described in [16,17]. In [18], the authors used a genetic algorithm to solve a network design problem in an urban area in a multi-criteria way. Aspects such as network layout link capacity were optimized. Particle swarm optimization (PSO) was also used for optimization and an example of its application is described in [19,20]. An innovative heuristic method called the nest of apes heuristic (NOAH) is described in [21]. The method can be used to model specific transport issues by combining technical aspects of transport systems with human decision-making.

This paper applies one method inspired by nature, namely ant colony optimization (ACO) [22]. Using it, it is possible to solve optimization problems in transport. In [23], the authors used an improved ACO to solve a vehicle routing problem (VRP). The improved model can use the traffic resource allocation to achieve a comprehensive optimization of time, cost and the number of accidents. In [24], the authors used a modified ant colony optimization (MACO) to evaluate underground rail route proposals, so that routes with different parameters could be evaluated. In [25], an improved ant colony optimization was proposed that uses an augmented pheromone update strategy, called the ant weight strategy. In [26], on the other hand, a new path transfer strategy of ants and a new dynamic pheromone update strategy applicable to time-dependent networks were proposed. Based on these strategies, the improved ACO was given for solving VRP in a time-dependent network. In [27], an improved ACO was developed for path planning based on a weight matrix, whereby the algorithm avoids multiple selection of paths with lower weights. It is also possible, using modifications, to optimize the urban bus network based on existing bus routes [28].

The metaheuristic approach is very useful for solving difficult optimization problems in practice. In [29], the aim of the research was to optimize a transport system using three metaheuristic algorithms: a genetic algorithm, an ACO and an extended great deluge (EGD). Paper [30] critically reviews the applications of metaheuristics for solving the transit route network design problem (TRNDP). The paper concludes with identified gaps in research and opportunities for future research on the application of metaheuristic algorithms for solving the TRNDP. In terms of initialization, the proposed methods utilize specific criteria to capture the qualities of a desirable route set, namely short travel times and a high percentage of direct travel. The planners’ knowledge has been explicitly incorporated in certain methods, as well as an effort to more realistically address the problem. In [31], the transit frequency optimization problem was studied, which aims to determine the interval between consecutive buses for a set of public transport lines defined by their routes, i.e., sequences of stops and street sections. A metaheuristic was proposed to solve larger instances, the accuracy of which was estimated by comparison with exact results (if possible). Using the proposed model, optimal or near-optimal solutions can be calculated for the case associated with a real small city. Although the considered public transport system has 13 lines, an improvement of about 3% is obtained by using the model.

In order for any transport system to be an attractive choice for travelers, demands such as competitive travel time must be met. The faster it is possible to travel by a given means of transport, the more attractive it will be as a choice. This simple statement has got multiple deep-reaching consequences which involve various disciplines, e.g., urban planning, environmental issues or the health of citizens, which altogether can be described in a synthetic form, e.g., using a life quality index. The mobility of the population forms an important factor in life choices and depends on many factors and patterns like the anisotropy and centripetality of the city [32]. It also corresponds well with the fact that, for a system as dependent on outside factors such as an average tram network, another method should be applied—one that will take into account the most important external factors influencing its ability to adhere to the timetable. It is also worth noting that total journey times depend not only on the speed of the vehicles, but also on other factors like pedestrian routes connecting the stations, or distances between stations (‘door to door’ travel time).

The aim of this study is to reduce tram journey times using ACO. The demonstration of the practical application of this method, namely the optimization of tramline routes in terms of running times and, in the future, the possibility of creating an entire network on this basis, is the main contribution of the work and makes it unique.

## 2. Methodology

### 2.1. Ant Colony Optimization

The ant colony optimization (ACO) is a meta-heuristic method inspired by the behavior of ant colonies in nature, which are able to find the shortest paths between a food source and the anthill. Ants in nature have very poor vision, typically limited to only a few centimeters. Therefore, they communicate with each other using a chemical called a pheromone. The algorithm makes use of this mechanism, the amplification of pheromone traces, to efficiently solve complex optimization problems, such as route optimization.

The algorithm can be described mathematically as follows: An optimization problem is represented as a graph, where the nodes correspond to possible states or decisions and the edges represent possible transitions between states. In addition, each edge has its own weight, depending on the objective function adopted. The ants explore the graph in search of optimal solutions. Initially, the ants move randomly, choosing a path with probability $p_{ij}^{k}$, determined by the following formula:(1)$$p_{ij}^{k}=\frac{\left(\tau_{ij}^{\alpha }\right)\left(\eta_{ij}^{\beta }\right)}{\sum_{z\epsilon {allowed}_{i}}\left(\tau_{ij}^{\alpha }\right)\left(\eta_{ij}^{\beta }\right)}$$

$p_{ij}^{k}$—probability of a *k*-ant crossing through section *ij*;

$\tau_{ij}$—the amount of pheromones at section *ij;*

*η_ij_*—the attractiveness of section *ij*.

In transportation issues, *η_ij_* is determined by the following formula:(2)$$\eta_{ij}=\frac{1}{L}$$
where L is the total route to be taken.

The coefficient *α* is the parameter that increases $\tau_{ij}$. The larger *α* is, the more we “trust” the information left by other ants. It is assumed that *α* ≥ 0. The coefficient *β* is the parameter that increases *η_ij_.* The larger *β* is, the more we ‘trust’ our own experience. It is assumed that *β ≥ 1*. The user of the algorithm can freely control the parameters α and *β*. During migration, ants secrete a constant amount of pheromones for other ants to follow. The amount of pheromones left behind can be determined by the following formula:(3)$$△\tau_{ij}^{k}=\left\{\begin{array}{c} Q/L_{k}\\ 0 \end{array}\right.$$
where:

Q—a fixed value indicating the amount of pheromone;

$L_{k}$—the length of route travelled by *k*-ant.

If the *k* ant passes through segment *ij*, the value of the coefficient ${∆\tau }_{ij}^{k}$ takes the value $Q/L_{k}$. In other cases, where the ant has not walked the path and has left no pheromones, the coefficient ${∆\tau }_{ij}^{k}$ takes the value 0. When determining the route, it is necessary to update the amount of pheromones. The pheromone updating process can be divided into two parts. The first part is the reduction of the pheromones located along section *ij* due to the natural evaporation of the pheromone. The second part is the increase in pheromones due to the following ants crossing in the next iteration. The amount of pheromones located along section *ij* after the update can be described by the following formula:(4)$${\tau^{\prime }}_{ij}=\left(1-\rho \right)\tau_{ij}+\sum_{k}∆\tau_{ij}^{k}$$
where:

${\tau^{\prime }}_{ij}$—the existing amount of pheromones at section ij;

${∆\tau }_{ij}^{k}$—the amount of pheromones left by *k* ant in the next iteration;

ρ—the evaporation coefficient of the pheromones.

The evaporation coefficient can take values from 0 to 1. When *ρ =* 1, then the pheromones evaporate completely after each iteration. Conversely, when *ρ =* 0, the pheromones evaporate completely. Adopting extreme values for the evaporation coefficient may adversely affect the obtained results and may increase the calculation time considerably. The pheromones left behind by the ants affect the attractiveness of the paths. Therefore, the more ants move along the path, the more attractive it becomes to subsequent ants. This affects the probability that the next ant leaving the nest will choose the path.

In the basic version of the algorithm, the given objective function for artificial ants is distance, i.e., the objective is to find the shortest path between the starting and ending points. This is not a necessary condition, and the objective function can be set differently. In this paper, the preset objective function is time; that is, the fastest route between the preset start and end points is searched for. As a result, the algorithm calculates the time it takes for ants to travel the route. ACO is based on a time matrix as follows:(5)$$T=\left[\begin{array}{ccc} t_{11} & \cdots & t_{1j}\\ \vdots & \ddots & \vdots \\ t_{i1} & \cdots & t_{ij} \end{array}\right]$$

The coefficient $t_{ij}$ determines the time it takes to cover the distance *ij*.

### 2.2. Determination of Travel Times

In this paper, the route of a tram on a selected line is optimized using ACO. Three objective functions are analyzed in the work: scheduled time, real time and theoretical time. To make the process of selecting the optimal route possible, time matrices are made for the entire tram network. Schedule time is determined based on the timetable available on the operator’s website (MPK Wrocław). Real time is determined on the basis of our own measurements. Furthermore, a database of each MPK Wrocław bus and tram location was created, based on February 2017. As described in [33], altogether, 61.3 million datapoints were collected, which were used to create spatio-temporal profiles of the mass transit routes. In order to perform future validation of the method proposed in this paper, another campaign of datapoint collection needs to be performed, then processed to eliminate the unwanted influence of changes and evolution of the route network. This real-time data may then be used to validate modifications resulting from the ant colony method.

For each section between stops, a minimum of three time measurements were made, from which the arithmetic mean is calculated. The full database with the measurements taken is available by contacting the authors. The theoretical time is defined as the time in which the tram could cover the section between stops *ij* and is given by the formula:(6)$$t_{t,ij}=\frac{L_{ij}}{V_{ij}}+t_{e}+t_{A}+t_{B}$$
where:

$t_{t,ij}$—theoretical running time between stops *ij*;

$L_{ij}$—distance between stops *ij*;

$V_{ij}$—average speed between stops *ij*;

$t_{e}$—passenger exchange time;

$t_{A}$—time increase for tram acceleration;

$t_{B}$—time increase for tram braking.

Passenger exchange times for the entire network are determined based on measurements made by the authors in the course of making measurements of the real driving times of trams. The measurements included 500 records, from which the average passenger exchange time interval is determined. As a result, the study assumed that passenger exchange time is a random variable with a range of 5–15 s. The average speed between stops depends on the distance between stops. The value of $V_{ij}$ is based on the Polish Standard WR-D-43-3 and is as follows:30 km/h for distances between stops up to 500 m;50 km/h for distances between stops between 500 and 700 m;70 km/h for distances between stops greater than 700 m.

The acceleration and braking time of the streetcar is added to the total theoretical time to account for the additional time loss involved. The values of $t_{A}$ and $t_{B}$ are defined by the following formulas:(7)$$t_{A}=\frac{V_{ij}}{2a_{r}}$$
(8)$$t_{B}=\frac{V_{ij}}{2a_{h}}$$
where:

$a_{r}$—tram acceleration value;

$a_{h}$—tram braking value.

## 3. Case Study

The study is carried out in one of the largest agglomerations in Poland. Wrocław is a city located in southwestern Poland in Lower Silesia. The city has a population of around 650,000 and about 1 million in the agglomeration. The tram network is very extensive with 190 km of single track. Its origins date back as far as the 19th century, when a horse-drawn tramway operated in the city. Due to its long history, the layout of the network is not regular, especially in the center. However, the radial layout of the network is clearly marked. The current layout of the network is shown in Figure 3. The network is shown using sections and nodes, with nodes indicating the location of stops. The red circle marks the city center. Such a model allows the creation of a timing matrix, which forms the basis of the ACO operation. The complex network layout is a good basis for carrying out research into improving tram journey times. This makes ACO especially suited for the task of tramway network optimization, as it remains independent of factors that influence the variability of time spent at a station and between the stops when programmed to do so. Regardless of the methods described in Chapter 1, external influences remain and will vary the results; for example, boarding time is a function of parameters like the horizontal and vertical gap between platform edge and step of the vehicle, door width, percentage of door openings in a side wall of the tram, passenger exchange on that particular stop, weather or time of the day. For the time being, authors are unable to propose a holistic, 100% verifiable model that would include the influence of all external factors.

The optimization process is the same for each tram line. Therefore, in this paper the authors will present the entire process for only one line. A line is chosen that has a cross-city route, i.e., through a region with the most complex network layout. Line No. 13 between the Olympic Stadium (node 72) and Nowy Dwór (node 170) is chosen. Its route is shown in Figure 4. The line crosses the most important traffic generators in the city and is an influential part of the overall network. It is therefore of significance that, following the optimization process, the line does not lose its importance and continues to pass through important points in the city.

For the route analyzed, alternative routes are calculated using ACO, in which the objective functions are schedule time, real time and theoretical time. The optimum scheduling time is determined by the timetable available on the operator’s website. The optimal real time is determined based on measurements taken by the authors of the paper. The time is calculated in seconds. The times are read between consecutive stops, from the opening of the tram doors at the previous stop to the opening of the doors at the next stop. This measurement allows the total travel time between stops to be determined. The optimum theoretical time is a mathematical model that determines how long a tram would travel under certain specified conditions, described in Equation (6). The ACO algorithm is implemented in the Python programming language, into which the timing matrix is introduced. Importantly, the start and end points remain unchanged and are defined at the Python code stage. The following parameters are assumed for the ACO calculation: number of ants, 300; number of iterations, 300; and evaporation factor, 0.5. The results from the study are shown in the next section.

## 4. Results

The obtained results are summarized in Table 1. To compare the results, all types of times are determined for all routes. The travel speed and route length are also determined. In addition, the obtained results are marked on the tram network, as shown in Figure 5. In the first place, it can be seen that the differences in the routes are visible in the downtown area, the place with the most irregular tram network. A different route is obtained for each variant. The differences in the obtained results vary according to the type of time. The lowest values are obtained for theoretical time and the highest for real time. The best theoretical time is 1479 s (24.65 min) and this converts into a driving speed of 30.91 km/h. The best scheduled time is 2340 s (39 min) and this results in a driving speed of 18.88 km/h. The best real time is 2645 s (44.08 min) and this gives a driving speed of 16.74 km/h. Comparing this final result with the current schedule of 44 min, it can be seen that a time saving of 17.6 min or 67% is possible with the current route and 19.35 min or 78% for the route with the best obtained theoretical time (alternative route 1).

## 5. Discussion

In this paper, the authors consider a nature-inspired tool to reduce tram journey times. Using ACO, the best theoretical, scheduled and real times are determined for a selected tram route on one of the largest tram networks in Poland. In fact, looking at Figure 5, it can be seen that all the obtained alternative routes have a different circuit than the current route, while keeping the same start and end point. The difference is especially noticeable between the current route and alternative routes 2 and 3. In these cases, in the downtown area, the routes have a completely different circuit. Importantly, the obtained routes meet the basic condition and pass through major locations in the city and do not change the nature of the streetcar line.

As stated in the previous paragraph, time savings of 7% or 13% may not seem impressive at a first glance. However, the effect of scale must be considered here. It is calculated that an average US citizen spends 51 h in a traffic jam every year, which is estimated at a cost of $869 [https://inrix.com/scorecard/, accessed on 21 June 2024]. Therefore, the cost of a wasted passenger-hour is roughly $17. Taking a time saving of 7% across 44 min, twice a day, over 220 working days a year, corresponds to approx. 1320 min in a year, which equals 22 h. This result is multiplied by the 200 million Wrocław network passengers, giving a total of 4.4 billion hours in a year or a grand total of $74.8 billion. Of course, the above calculation is extreme and overly simplified, as time savings or cost savings depend on multiple factors, such as the total distance commuted daily. Furthermore, all modes of transport alternative to the tramway include wasted time; consequently, not every minute saved may be counted as productive. This example is intended only to show how easily single-minute savings, often overlooked by both decision-makers and general public, accumulate into a substantial amount of time or money.

Many operators strive to achieve a certain operational speed. In the case of Wrocław, the expected operational speed of the trams is 20 km/h. Comparing the obtained result with the current schedule, which corresponds to an operational speed of 16.76 km/h, it can be seen that the expected speed is not achieved. Among the alternative routes, the closest to the expected schedule speed is alternative route 2 (18.88 km/h). The highest speeds are obtained for the theoretical time, which are significantly above the 20 km/h value. The highest theoretical speed is 30.91 km/h. Even for the current route, the speed is 27.94 km/h. In contrast, obtaining the theoretical time, and thus obtaining the theoretical speed, under real conditions on the network is a very difficult task. However, changes can be made on the network; for example, by giving priority at junctions, which will increase the speed of travel and, as a result, at least achieve the desired operational speed. Furthermore, the method described in this paper may serve as a tool to indicate sections and nodes of the network which, when improved, will bring the largest time savings for all of the lines running through them.

The obtained results allow us to proceed to the optimization and selection of the best route. First, it is necessary to consider which of the determined times to take as the leading one. It is best to take the time that has the lowest value, which is the theoretical time. Comparing the current route with the alternative routes in Table 1, the obtained theoretical time for the current route (26.4 min) is 7% worse than the best theoretical time (alternative route 1; 24.65 min). Unfortunately, taking the theoretical time as the optimal time is not the best solution due to the limitations of this method, which are described in the next section. With the tentative acceptance of the scheduling time as the optimal one, this time (44 min) is 13% worse for the current route than for the best one (alternative route 2; 39 min). The problem with this time is the way it is determined, and for peak hours it can cause significant delays. Practical experience teaches that time reserves of various natures need to be included in timetables; otherwise, the slightest delay can cause an avalanche of consequences.

According to the authors, it is best to optimize the route in terms of real time, as it best reflects traffic conditions along the route and implicitly takes into account issues related to passenger exchanges, infrastructure conditions, etc. The obtained real time for the current route (44.57 min) is close to the best obtained real time and is only 1% worse (alternative route 3; 44.08 min). As the difference between the times is small, the remaining times can be used for route selection. The theoretical times between the current route and alternative route 3 are small, so the schedule time is crucial in this case. The difference between these times is 4 min.

Finally, after analyzing the obtained results, route optimization is performed and alternative route 3 is selected as the best circuit on this route. The optimization procedure shown can be carried out for any existing tramway line. In this way, optimization of the entire network can be carried out.

## 6. Conclusions

The research methodology used in this paper allowed the verification of tram route variants in terms of travel time. The results made it possible to compare the variants, identify the differences between them and select the best option. The proposed tool using a nature-inspired algorithm made it possible to obtain better travel times for the analyzed tram line, which confirms the effectiveness of the tool. The demonstration of the practical application of the method, namely the optimization of tram line routes in terms of travel times and, in the future, the possibility to create an entire network on this basis, is the main contribution of the work and makes it original. Importantly, the tool can be applied to any mode of public transport, which makes it a universal tool. It allows a high possibility of modification and adaptation to the problem under consideration.

The results indicate that there are issues that need further consideration in future research. Challenges present in the proposed method include, but are not limited to, insufficient control over the results: ACO has tendency to propose fastest routes, yet in a real city two more factors have to be respected, namely serving the locations surrounding the slower routes and preventing the overcrowding of optimal rotes with multiple lines, which obviously would cause an increased transfer of disturbances between the vehicles. Furthermore, some mass transit services are held as a social service, regardless of their performance or economic return. In conclusion, to obtain the best results, ACO should be used to optimize the routes of all lines simultaneously.

Another weakness of ACO is that it remains insensitive to local improvements in infrastructure, like an increased platform edge height, a higher priority on a crossing, new pedestrian signals or a turnout that can be driven over at higher speed. All such actions, regardless if applied on the outer branches or in central bottlenecks of the network, will call for another round of calculations. The authors also find it difficult to directly transfer data like track quality or condition, with their influence on allowed speed, directly to the proposed method.

A problematic issue in the paper is the simplification of certain factors when determining the theoretical time. These factors are the speed of the trams and the passenger exchange time. These are important elements in determining the total journey time. The speed of the trams depends on many external factors such as traffic conditions, the condition of the infrastructure, the impact of speed limits on curves, pedestrian crossings and the skills of the driver. The same is true for passenger exchange times. It depends largely on the location of the stops and the time of day. Due to the randomness of these factors, it is difficult to determine them in a discrete way. This presents a challenge that will be considered in future research. To fully validate the method, a database similar to the one collected in 2017 needs to be created and thoroughly compared with the previous one. This will point out the nodes and edges in the network that are theoretically optimal but are practically prone to delays. Future studies would also need to compare the proposed method with another optimization method in order to better verify the results obtained.

## Figures and Tables

**Figure 1 sensors-24-06226-f001:**
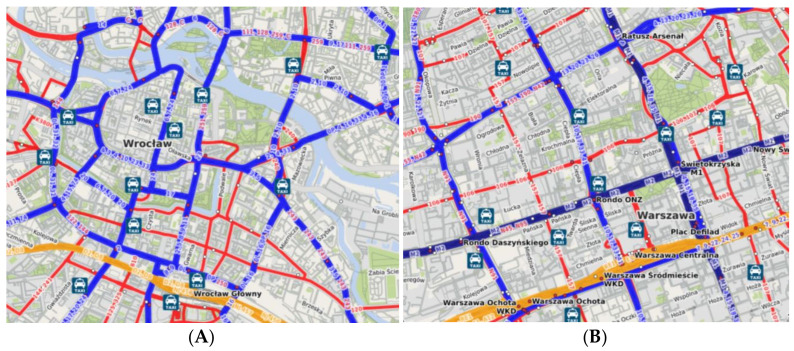
Tram network (blue) in center of Wrocław, Poland (**A**) and Warsaw (**B**). Source: öpnvkarte.de.

**Figure 2 sensors-24-06226-f002:**
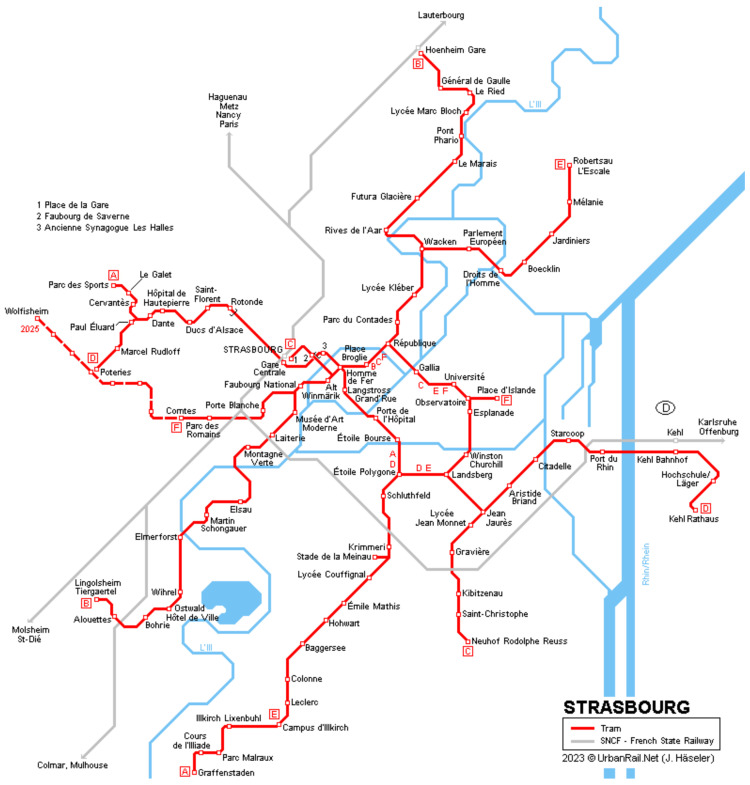
Tram network in Strasbourg is roughly half the size of Wrocław’s, yet it relies on ‘one route, one line’ principle on the outskirts and contains only one track crossing, at central point ‘Homme de Fer’. Source: UrbanRail.net, by J. Häseler.

**Figure 3 sensors-24-06226-f003:**
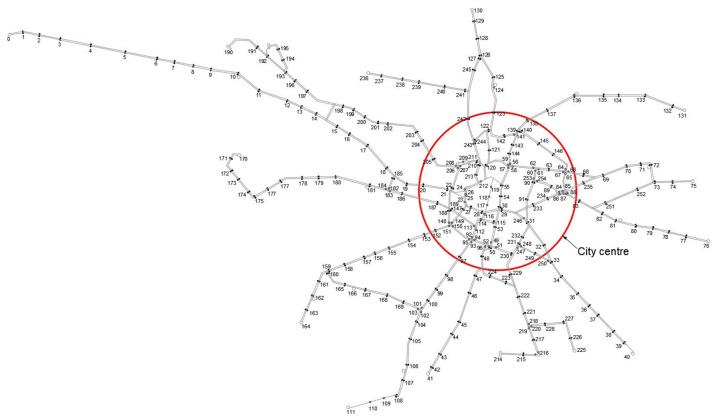
Layout of the tramway network in Wrocław.

**Figure 4 sensors-24-06226-f004:**
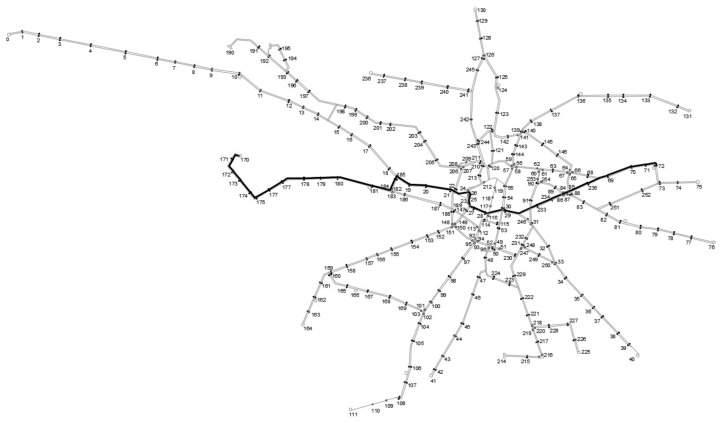
Current route of the considered tramway line 13.

**Figure 5 sensors-24-06226-f005:**
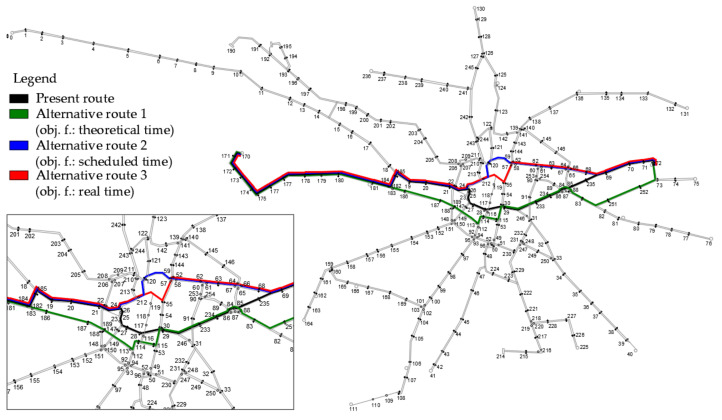
Routes received traced on the tram network.

**Table 1 sensors-24-06226-t001:** Summary of results.

	Present Route	Alternative Route 1 (Objective Function: Theoretical Time)	Alternative Route 2 (Objective Function: Scheduled Time)	Alternative Route 3 (Objective Function: Real Time)
Number of stops	29	26	29	29
Length [m]	12,293	12,700	12,274	12,302
Theoretical time [s]	1584 (26.4 min)	1479 (24.65 min)	1612 (26.87 min)	1617 (26.95 min)
Scheduled time [s]	2640 (44 min)	2640 (44 min)	2340 (39 min)	2400 (40 min)
Real time (average) [s]	2674 (44.57 min)	2918 (48.63 min)	2676 (44.6 min)	2645 (44.08 min)
Theoretical speed [km/h]	27.94	30.91	27.41	27.39
Scheduled speed [km/h]	16.76	17.32	18.88	18.45
Real speed [km/h]	16.55	15.67	16.51	16.74

## Data Availability

Data is contained within the article.

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
