# Peer review of "Applying Ant Colony Optimization to Reduce Tram Journey Times"

_sensors, 2024, doi:10.3390/s24196226_

Round 1

Reviewer 1 Report

Comments and Suggestions for Authors

1. In the abstract, research background and significance are poorly written. The main innovations need to be highlighted, and the logic of language expression is not well.

2. The current literature review is shallow and requires a deeper review to adequately support the study. Since the paper contributes to the metaheuristic algorithms, the authors should provide a detailed categorization of metaheuristic algorithms, as well as their applications in solving real-world problems.

3. In the introduction, the main contribution and originality should be explained in more detail.

4. It is recommended to add specific parameter settings for the ACO algorithm.

5. The authors should incorporate the best-so-far proposed variants of the ACO and add some experiments of well-known metaheuristic algorithms and other literatures to verify the advantages of the proposed algorithm, the effectiveness of this method needs to be further proved in Section 4.

6. The authors should recheck the manuscript and correct the typos.

7. Limitations of the proposed method are not discussed.

Comments on the Quality of English Language

The authors should recheck the manuscript and correct the typos.

Author Response

Thank you for your review. The responses to it are in the annex.

Reviewer 2 Report

Comments and Suggestions for Authors

The paper proposes a solution to the problem of finding optimal tram routes based on ant colony optimization method.  The objective function is the travel time by analyzing the planned, actual and theoretical time.

I believe that the authors have put a lot of effort into demonstrating their proposed optimization method for a selected tram route of one of the largest tram networks in Poland, as well as using the comparison of this route with the proposed three alternative routes obtained by optimization with different objective functions.

My remarks are:

1. The summary states that ".. The routes are then compared to each other to select the optimal solution..." The analysis and discussion of the obtained results do not comment on the choice of an optimal solution, although 3 alternatives are proposed. I think that from this point of view, a more thorough analysis of the obtained results is necessary, and not just their comparison in percentages.

2. It would be appropriate to compare the proposed method with another optimization method suitable for the given task, so that the obtained results are more convincing and the discussion is more thorough and useful for the readers.

Author Response

(The authors gave the same response as above.)
